# Incidence of HBV Reactivation in Psoriasis Patients Undergoing Cytokine Inhibitor Therapy: A Single-Center Study and Systematic Review with a Meta-Analysis

**DOI:** 10.3390/v17010042

**Published:** 2024-12-30

**Authors:** Meng Hsuan Kuo, Ping-Hung Ko, Sz-Tsan Wang, Chih-Wei Tseng

**Affiliations:** 1Department of Pharmacy, Dalin Tzu Chi Hospital, Buddhist Tzu Chi Medical Foundation, Chia-Yi 622, Taiwan; df441865@tzuchi.com.tw; 2School of Medicine, Tzuchi University, Hualien 970, Taiwan; a0911730070@gmail.com (P.-H.K.); curtis210888@gmail.com (S.-T.W.); 3Division of Gastroenterology, Department of Internal Medicine, Dalin Tzu Chi Hospital, Buddhist Tzu Chi Medical Foundation, Chia-Yi 622, Taiwan; 4Division of Rheumatology, Department of Internal Medicine, Dalin Tzu Chi Hospital, Buddhist Tzu Chi Medical Foundation, Chia-Yi 622, Taiwan

**Keywords:** cytokine inhibitors, HBV reactivation, interleukin-17, interleukin-23, psoriasis

## Abstract

Background: Psoriasis patients who are seropositive for hepatitis B surface antigen (HBsAg) or hepatitis B core antibody (HBcAb) face an elevated risk of hepatitis B virus reactivation (HBVr) when treated with cytokine inhibitors. This study aims to elucidate the risk in this population. Methods: A retrospective chart review was conducted to assess the risk of HBVr in 73 psoriasis patients treated with cytokine inhibitors from 2013 to 2023. Additionally, a systematic review and meta-analysis were performed, pooling data from 10 studies (including our cohort) and adhering to PRISMA guidelines. Statistical heterogeneity was assessed using the *I*^2^ statistic, and pooled proportions were calculated using a random effects model. Results: No HBVr cases were observed among the 11 HBsAg+ patients in the cohort. However, two of the sixty-two (3.2%) HBsAg−/HBcAb+ patients experienced reactivation during therapy, with outcomes ranging from spontaneous recovery in one case to death from hepatic failure despite antiviral treatment in the other. The meta-analysis, pooling data from 10 studies, revealed a reactivation rate of 21.2% (95% CI: 9.4–41.0%) in HBsAg+ patients without prophylaxis and 4.4% (95% CI: 2.2–8.7%) in HBsAg−/HBcAb+ patients. Conclusion: Antiviral prophylaxis is essential for HBsAg+ patients receiving cytokine inhibitors, given the high risk of reactivation. Despite the lower risk for HBsAg−/HBcAb+ patients, the potential severity of outcomes demands careful monitoring and timely action.

## 1. Introduction

Biologic agents, particularly those targeting tumor necrosis factor-alpha (TNF-α), interleukin-17 (IL-17), and interleukin-23 (IL-23), have revolutionized the management of moderate to severe psoriasis [1,2]. These agents are highly effective in reducing disease severity, thereby enhancing skin clearance and improving quality of life [3]. However, while these therapies are beneficial, they also suppress the immune system and increase the risk of infections, including the reactivation of latent infections, such as hepatitis B [4]. The reactivation of hepatitis B virus (HBV) can lead to severe liver damage and even fatal outcomes, representing a significant concern for patients with past or current HBV infections. HBV infection among psoriasis patients ranges from 0.45% to 5.6%, influenced by geographic and population characteristics [5,6]. In HBV-endemic regions, such as Taiwan, the risk of reactivation represents a significant clinical challenge [7].

The risk of HBV reactivation (HBVr) and the necessity for antiviral prophylaxis are influenced by both the type of immunosuppressive drugs used and the patient’s HBV serostatus [8,9]. Among the biological agents used for psoriasis, the risk of HBVr with TNF-α inhibitors is well documented. In clinical practice, antiviral prophylaxis is required for hepatitis B surface antigen-positive (HBsAg+) patients due to reactivation rates ranging from 14 to 63% [8]. In contrast, patients with resolved HBV infection (HBsAg−/hepatitis B core antibody positive [HBcAb+]) have a lower risk, estimated at 1%, and typically only require monitoring [10]. Compared with TNF-α inhibitors, the evidence of the HBVr risk in psoriasis patients receiving cytokine inhibitors (IL-12/23 inhibitor, IL-17 inhibitors, IL-23 inhibitor) was weak [8].

Although research has focused on psoriasis patients receiving cytokine inhibitors, most studies have been limited by small sample sizes [4,11,12,13,14,15,16,17,18]. Despite the overall limited number of cases, some meta-analyses have sought to address this knowledge gap by pooling data from these studies [10,19,20]. The pooled results demonstrated a reactivation risk of 25–36% for HBsAg+ patients and 3–5% for HBsAg−/HBcAb+ patients [10,19]. To accurately determine the true reactivation rate, additional reports and updated meta-analyses are necessary to further refine our understanding.

The purpose of this study is to evaluate the risk of HBVr in psoriasis patients receiving cytokine inhibitors. To achieve this, we conducted a retrospective review of psoriasis patients in our hospital. Due to the limited number of cases in our single-center study, pooling data from multiple studies allowed us to increase the statistical power and generalizability of our findings. To achieve this, additionally, a comprehensive analysis was performed through a systematic review and meta-analysis, pooling data from our cohort and existing research.

## 2. Materials and Methods

This retrospective study analyzed adult patients with moderate-to-severe psoriasis receiving cytokine inhibitors at Dalin Tzu Chi Hospital, Buddhist Tzu Chi Medical Foundation in southern Taiwan from January 2013 to December 2023. The inclusion criteria focused on adult patients, while the exclusion criteria removed individuals younger than 20, those without complete HBV serum marker records, and those not previously exposed to HBV (HBsAg−/HBcAb−). This research adhered to the Helsinki Declaration and received approval from the Ethics Committee of Dalin Tzu Chi General Hospital (approval number B11303018). Due to the retrospective design and anonymization of patient data, informed consent was waived.

The medical records of patients were examined retrospectively to gather data including demographic details (gender and age), clinical characteristics of psoriasis, disease duration, initial laboratory findings, and historical treatment approaches for psoriasis as well as hepatitis B and C virus (HBV and HCV) infections. The clinical evaluation focused on HBV serological markers (HBsAg, HBcAb, HBsAb, and HBV DNA), levels of alanine aminotransferase (ALT), existing comorbidities, ongoing medications, and incidents of HBVr. ALT values were checked every three months, and assays for HBsAg and HBV DNA were performed as clinically indicated. Additionally, comprehensive data regarding the course of HBVr were recorded for affected patients.

Reimbursement for biologic treatments for the treatment of moderate-to-severe psoriasis in Taiwan is governed by the policies of the National Health Insurance (NHI). Eligibility for biologic therapy requires that patients have previously experienced unsuccessful outcomes with phototherapy and have utilized at least two traditional systemic agents, such as methotrexate, acitretin, or cyclosporine [21]. Available biologic options include tumor necrosis factor-alpha inhibitors (anti-TNF-α agents: adalimumab and etanercept), the anti-IL 12/23 biologic (ustekinumab), IL-17 inhibitors (secukinumab, brodalumab, and ixekizumab), and the anti-IL 23 biologic (risankizumab). It is noteworthy that the NHI does not routinely cover prophylactic antiviral therapy for patients undergoing biologic treatments. Patients with chronic hepatitis B are monitored by hepatologists and may receive antiviral treatment for hepatitis based on established reimbursement criteria [22]. Decisions to provide antiviral prophylaxis to patients who fall outside these reimbursement parameters are made through discussions between the patients and their physicians.

In this study, the primary outcome was HBVr. For HBsAg+ patients, HBVr is characterized by any of the following criteria: (1) an increase in HBV DNA exceeding 2 log10 from baseline levels, (2) an HBV DNA level above 3 log10 in cases previously undetectable, or (3) an absolute HBV DNA level over 4 log10 if the baseline data are unavailable. In HBsAg−/HBcAb+ patients, reactivation is indicated by either a detectable increase in HBV DNA from an undetectable state or HBsAg seroreversion. Hepatitis flare-up is defined as an ALT increase to at least three times the baseline value, exceeding 100 U/L. The potential for cytokine inhibitors to induce HBVr was assessed based on their use at either the end of the follow-up period or at the time of HBVr detection.

ALT levels were measured using automated biochemical analyzers (Roche Analytics; Roche Professional Diagnostics, Penzberg, Germany). The cut-off value for the serum ALT level was 40 IU/L. Serological markers for HBV (HBsAg, HBcAb, and HBsAb) were assessed using chemiluminescence immunoassays (Abbott ARCHITECT i2000, IED: Houston, TX, USA). Serum HBV DNA levels were measured using the COBAS^®^ HBV quantitative nucleic acid test on the COBAS^®^ 4800 System (Roche Diagnostics, Indianapolis, IN, USA), with a lower limit of quantification of 5 IU/mL. These evaluations were conducted at regular intervals in accordance with clinical guidelines.

Statistical analysis was performed using MedCalc Version 22.002. Continuous variables were reported as median with range, while categorical variables were presented as numbers with percentages (%). The Mann–Whitney U test was used to compare continuous variables, and the chi-square test or Fisher’s exact test was used for categorical data, as appropriate. A *p*-value of <0.05 was considered statistically significant.

This study synthesizes findings from existing research and recent results to conduct a meta-analysis on HBVr among psoriasis patients treated with cytokine inhibitors. Adhering to a registered protocol on INPLASY (registration number INPLASY202490005), this analysis complies with the Preferred Reporting Items for Systematic Reviews and Meta-Analyses (PRISMA) guidelines, detailed in Appendix A. Independent literature searches were conducted by two authors, MHK and CWT, across databases, including PubMed, Embase, Web of Science, and the Cochrane Central Register of Controlled Trials, spanning from their inception until 2 July 2024. The methodologies, including the search strategy and key terms, are systematically outlined in Appendix A. Any arising discrepancies were collaboratively resolved through detailed discussions.

This meta-analysis included both observational studies and randomized controlled trials, applying no restrictions based on language. Studies were excluded if they had fewer than five cases, did not report HBV status, or involved overlapping populations with smaller subsets. We evaluated the risk of bias within observational studies using the Newcastle–Ottawa Scale (NOS).

A random effects meta-analysis of single proportions was conducted to estimate the pooled rate of HBVr, and statistical heterogeneity among the studies was assessed using the *I*^2^ statistic. Subgroup analyses were stratified by type of cytokine inhibitors (IL-12/23 inhibitors, IL-17 inhibitors, IL-23 inhibitors), geographic regions (Asian versus non-Asian), and HBsAb status (positive and negative). To mitigate the impact of small sample sizes, subgroups containing only a single study were excluded from the analysis. Publication bias was planned to be evaluated through funnel plot asymmetry, contingent on the inclusion of ten or more studies for any given outcome. To ensure the robustness of our findings, sensitivity analyses were performed using the one-study removal method. Statistical significance was established at a *p*-value of less than 0.05, with all computations executed using Comprehensive Meta-Analysis software version 4.0.

## 3. Results

### 3.1. Characteristics of the Study Participants

Figure 1 shows the distribution of study participants, starting with 160 psoriasis patients receiving cytokine inhibitors. Exclusions were made for those under 20 years of age (*n* = 3), those with incomplete HBV serum data (*n* = 11), and those without prior HBV exposure (HBsAg−/HBcAb-) (*n* = 73), leaving 73 patients for analysis. Among them, eleven tested HBsAg+, with six on antiviral prophylaxis; none of these patients had HBVr. Meanwhile, among the sixty-two patients who were HBsAg−/HBcAb+, two (3.2%) experienced HBVr during cytokine inhibitor therapy.

Table 1 presents the demographic and clinical characteristics of the study participants. The median age of the cohort was 51 years, ranging from 28 to 87 years, with a predominance of males (63%, *n* = 45). The median duration of disease follow up was 8.1 years, with a range from 0.3 to 23.6 years. Over the course of this study, most patients (84%, *n* = 61) were treated with an IL-17 inhibitor, while smaller proportions were treated with IL-12/23 inhibitors (13%, *n* = 9) and IL-23 inhibitors (4%, *n* = 3). Regarding prior therapies, most patients had been treated with methotrexate (90%) and cyclosporine (39%). Additionally, a significant subset (35%, *n* = 25) had a history of TNF-alpha inhibitor usage, with adalimumab being the most administered agent (22%, *n* = 16). Among patients with HBsAg−/HBcAb+, 68% (42/62) were HBsAb+.

Baseline characteristics were also compared between HBsAg+ and HBsAg−/HBcAb+ groups, as indicated in Table 1. Statistically significant differences were observed in baseline HBsAb status (*p* < 0.001) and HBV prophylaxis rates (*p* < 0.001). Other variables did not show statistically significant differences between the groups.

### 3.2. Clinical Features in Patients with HBV

Appendix A details the clinical data of two HBsAg−/HBcAb+ patients who experienced HBVr. The first case involves a female patient who was HBsAg− and HBcAb+ prior to treatment, with an undetectable HBV DNA titer and HBsAb levels of <10 IU/mL. She had a long-standing diagnosis of psoriasis, which was effectively managed for 22 years using methotrexate and etanercept. After discontinuing etanercept for four months, she experienced a flare-up of psoriasis characterized by scaling and subsequently commenced therapy with brodalumab. Following the administration of three doses of brodalumab (3 months), her initially undetectable HBV DNA levels became detectable. Despite not receiving antiviral therapy, her viral load returned to undetectable levels within four months, allowing her treatment with brodalumab to continue without interruption.

The second case involved a male patient who was also HBsAg− and HBcAb+, with an undetectable pre-treatment HBV DNA titer. Data on his HBsAb levels were unavailable. He was initially treated with seven doses of secukinumab and was transitioned to ustekinumab due to inadequate response. After five doses of ustekinumab (9 months), the patient was presented to the emergency department with general weakness. At this time, his HBV DNA levels had escalated to 2.8 × 10^7^ IU/mL, coinciding with a hepatitis flare-up indicated by an ALT level of 165 U/L and severe jaundice, with total bilirubin at 18.1 mg/dL. Despite the immediate initiation of entecavir, the patient succumbed to hepatic failure 17 weeks later.

### 3.3. Systematic Review and Meta-Analysis

The study selection process is illustrated in the flow diagram (Appendix A). Characteristics of the 10 included studies are detailed in Appendix A [4,8,9,10,11,12,13,14,15]. Appendix A lists excluded studies and the reasons for their exclusion. Six articles provided information on the rate of HBVr in HBsAg+ patients and HBsAg−/HBcAb+ patients.

Five studies, encompassing 25 HBsAg+ patients who did not receive antiviral prophylaxis, reported the HBVr rate [4,8,9,10]. The pooled HBVr rate for HBsAg+ patients without antiviral prophylaxis was 21.2% (4/25, 95% CI: 9.4–41.0%; *I*^2^ = 0%), as shown in Figure 2A and Table 2. The median time from the initiation of cytokine inhibitor treatment to HBVr in this group was 5 months (range: 3–7 months). In contrast, among the 31 HBsAg+ patients who received antiviral prophylaxis, reported across six studies, there were no reactivation events (0/31) [8,9,10,11,12]. Additionally, ten studies included a total of 218 HBsAg−/HBcAb+ patients not on antiviral prophylaxis. The pooled rate of HBVr in this group was 4.4% (95% CI: 2.2–8.7%; *I*^2^ = 0%), as illustrated in Figure 2B and detailed in Table 2 [4,8,9,10,11,12,13,14,15]. The median time to reactivation from the start of cytokine inhibitor therapy in these patients was 11 months (range: 2–12 months).

The subgroup analysis performed on HBsAg−/HBcAb+ patients revealed no significant heterogeneity in HBVr risk according to the type of cytokine inhibitors used (IL-12/23 inhibitor, IL-17 inhibitors, IL-23 inhibitor), the study regions (Asian vs. non-Asian), and HBsAb status (positive vs. negative), as detailed in Table 2. This indicates a consistent risk profile across various subgroups and therapeutic modalities.

The subgroup analysis, detailed in Appendix A, investigated the influence of HBsAb (hepatitis B surface antibody) on the risk of HBVr among HBsAg−/HBcAb+ patients. The findings revealed no significant differences in reactivation risk between patients with and without HBsAb, with a risk difference of 0.04% (95% CI: −0.08 to 0.16%). Comparable outcomes were observed in additional subgroup analyses that considered variations in cytokine inhibitor type and geographic region.

The symmetry displayed in the funnel plot, which assessed the prevalence of HBVr, indicates a lack of publication bias, corroborated by Egger’s test (*p* = 0.56) (Appendix A). Furthermore, sensitivity analyses using the one-study removal method consistently supported these results (Appendix A). The consistency in findings, despite the removal of any individual study, underscores the robustness of the pooled effect sizes from this meta-analysis.

## 4. Discussion

This retrospective study with a meta-analysis demonstrated a pooled reactivation rate of up to 21.2% in HBsAg+ psoriasis patients receiving cytokine inhibitors. No HBVr occurred in HBsAg+ patients who received antiviral prophylaxis, underscoring the effectiveness of prophylactic treatment in this group. In contrast, among HBsAg−/HBcAb+ patients, the pooled reactivation risk though lower at 4.4%, remains clinically significant. Importantly, one of the two reactivation cases in our cohort resulted in a fatal outcome, emphasizing the potential severity of HBVr in HBsAg−/HBcAb+ patients.

In our retrospective cohort, no HBVr was observed among HBsAg+ psoriasis patients without antiviral prophylaxis (0/5), which is consistent with the findings of Lu et al. (0/3; treated with ustekinumab) [4] and Qin et al. (0/2; treated with secukinumab) [13]. In contrast, Chiu et al. and Ting et al. reported reactivation rates of 28.5% (2/7) [11] and 25% (2/8) [12] with ustekinumab treatment. Among those receiving antiviral prophylaxis, no cases of HBVr were observed [4,11,12,13]. The limited sample sizes and variability in these studies highlight the need for meta-analyses to accurately assess the true risk.

A meta-analysis conducted by Papatheodoridis et al. included six studies involving rheumatologic HBsAg+ patients treated with cytokine inhibitors [10]. Among patients without antiviral prophylaxis, the pooled HBVr rate was 36% (16/45; 95% CI: 23–50%; heterogeneity, *p* = 0.73). In contrast, no HBVr cases were observed in patients receiving prophylaxis (0/26). Similarly, a meta-analysis by Kuo et al. focused on psoriasis patients treated with cytokine inhibitors and analyzed three studies with 17 HBsAg+ patients, reporting an HBVr rate of 25% (95% CI: 10.4–49.7%) without antiviral prophylaxis [19]. Expanding on these findings, our updated meta-analysis incorporated five studies with 56 HBsAg+ patients. We observed a 21.2% reactivation rate (4/25; 95% CI: 9.4–41.0%) in patients without prophylaxis compared to no cases of reactivation in those receiving prophylaxis (0/31). These results strongly underscore the critical importance of prophylactic treatment in this high-risk population.

The reactivation rate in HBsAg−/HBcAb+ psoriasis patients was significantly lower compared to HBsAg+ individuals [19,20]. In our retrospective cohort, the HBVr rate was 3.2%. Papatheodoridis et al. analyzed eight studies involving 235 HBsAg−/HBcAb+ patients with rheumatic disease and revealed a reactivation rate of 3% (95% CI 1–6%; heterogeneity, *p* = 0.87) among cytokine inhibitor users [10]. Similarly, a meta-analysis by Kuo et al. comprising eight studies with 138 HBsAg−/HBcAb+ psoriasis patients reported a reactivation rate of 5.0% (95% CI 2.3–10.8%, *I*^2^ = 0%) [19]. Our updated meta-analysis with pooling data from 10 studies with 218 patients refines the risk estimate for HBVr in this group to 4.4% (95% CI: 2.2–8.7%; *I*^2^ = 0%). Although the risk is lower than in HBsAg+ patients, it persists, necessitating careful monitoring.

Although the HBVr rate was low in patients with HBsAg−/HBcAb+, the variability in clinical outcomes following reactivation warrants careful consideration. In our cohort, one patient succumbed to fatal hepatic failure despite receiving antiviral therapy. Previous studies have shown that HBVr-associated hepatitis occurred in only 1 out of 235 patients treated with cytokine inhibitors (pooled rate: 0%, 95% CI 0–3%; heterogeneity, *p* = 1.00) [10]. No cases of HBVr-related liver decompensation or death were observed among HBsAg+ or HBsAg−/HBcAb+ patients [10]. Nevertheless, the fatal case in this cohort underscores the potential unpredictability of HBVr outcomes, emphasizing the critical need for timely detection and early antiviral intervention.

Although the presence and concentration of anti-HBs antibodies are considered protective factors against HBVr in HBsAg−/HBcAb+ patients with rheumatic disease, 20–22 subgroup analyses revealed that these antibodies do not alter the incidence of HBVr following the use of cytokine inhibitors [23,24,25]. The consistent risk of HBVr across different cytokine inhibitors (IL-12/23, IL-17, IL-23 inhibitors) and geographic regions was also observed, suggesting that the reactivation risk in psoriasis patients is independent of these factors. The lack of heterogeneity in the meta-analysis results, coupled with the absence of publication bias, further reinforces the reliability of these findings. This consistency enhances the generalizability of this study’s conclusions, making them applicable across various clinical settings and populations.

This study has a few limitations worth noting. First, the small sample size in our single-center study is a challenge. To address this, we combined our cohort data with existing research through a systematic review and meta-analysis. This combined approach enhanced the statistical power of our findings, providing clinicians with more reliable insights into HBVr risks among psoriasis patients treated with cytokine inhibitors. Second, the retrospective, observational nature of this study might have led to some missing data. Infrequent monitoring of HBsAg and HBV DNA may have delayed the recognition of HBVr, making it difficult to use changes in HBV markers to predict reactivation early. To establish causality more clearly, randomized controlled trials would be essential. Third, our study focused solely on Asian patients, most of whom were treated with IL-17 inhibitors, which could limit how well these findings apply to patients on other cytokine inhibitors. However, by pooling data from available studies and performing subgroup analyses, we confirmed that our results are consistent across different groups. Future research should delve into the reactivation risks linked to newer cytokine inhibitors and emerging therapies to ensure thorough risk management.

## 5. Conclusions

Our research reveals the risk of HBVr in psoriasis patients on cytokine inhibitors, highlighting the high rate of HBVr and the effectiveness of antiviral prophylaxis in HBsAg+ individuals. Although HBsAg−/HBcAb+ patients face a lower risk, the potential for fatal outcomes stresses the need for timely detection and early antiviral intervention.

## Figures and Tables

**Figure 1 viruses-17-00042-f001:**
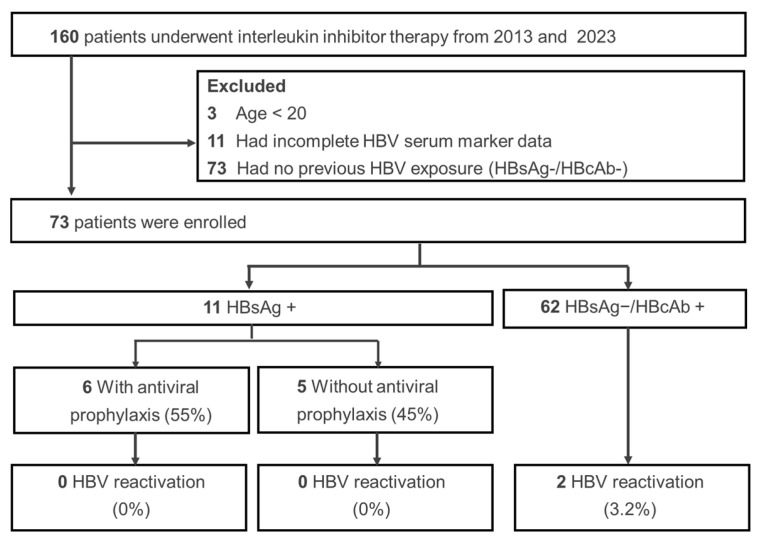
Flow diagram of study cohort characteristics.

**Figure 2 viruses-17-00042-f002:**
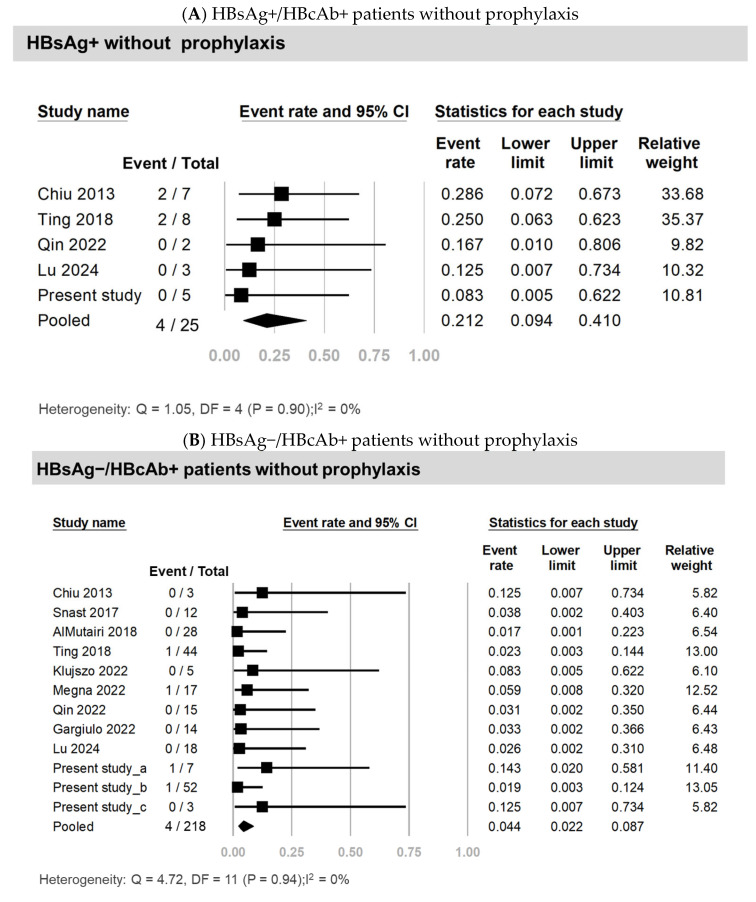
Pooled rates of HBVr in psoriasis patients treated with an interleukin inhibitor.

**Table 1 viruses-17-00042-t001:** Baseline patient characteristics.

Variables #	All (N = 72)	HBsAg+ (N = 11)	HBsAg−/HBcAb+ (N = 62)	*p*-Value ^&^
Age (years)	51 (28–87)	50 (28–80)	54 (34–87)	0.16
Sex, male, n (%)	45 (63%)	8 (73%)	37 (60%)	0.52
Baseline HBsAb+ (>10 mIU/mL) ^ʃ^	42 (58%)	0 (0%)	42 (68%)	<0.001
Baseline ALT, IU/mL	24 (8–87)	22 (8–65)	24 (8–87)	0.93
HCV positive, n (%)	5 (7%)	1 (9%)	4 (6%)	0.57
HBV prophylaxis	6 (8%)	6 (55%)	0 (0%)	<0.001
Disease follow-up time (year)	8.1 (0.3–23.6)	7.0 (2.3–23.3)	8.1 (0.3–23.6)	0.67
Time from cytokine inhibitor initiated (year)	2.7 (0.3–8.3)	2.8 (1.5–8.2)	2.6 (0.3–8.3)	0.15
Time from bDMARDs initiated (year)	5.0 (0.1–22.3)	5.3 (2.3–20.2)	5.0 (0.1–22.3)	0.69
Cytokine at the end of follow up				
IL-17 inhibitor	61 (84%)	9 (82%)	52 (84%)	1.00
IL-12/23 inhibitor	9 (13%)	2 (18%)	7 (11%)	0.62
IL-23 inhibitor	3 (4%)	0 (0%)	3 (5%)	1.00
Therapies before cytokine therapy				
Methotrexate, n (%)	65 (90%)	9 (82%)	56 (90%)	0.59
Cyclosporine, n (%)	28 (39%)	6 (55%)	22 (35%)	0.33
TNF-alfa				
Adalimumab, n (%)	16 (22%)	3 (27%)	13 (21%)	0.69
Etanercept, n (%)	9 (13%)	3 (27%)	6 (10%)	0.13
Golimumab, n (%)	11 (15%)	2 (18%)	9 (15%)	0.67
HBVr, n (%)	2 (2.7%)	0 (0%)	2 (3.2%)	1.00

# Continuous variants are presented as median (range); categorical variants are presented as n (%). ^&^ The *p*-values for comparisons using the Mann–Whitney U test for continuous variables and the chi-square test or Fisher’s exact test for categorical variables. ^ʃ^ missing data in baseline HBsAb+ (>10 mIU/mL): 2 in HBsAg+ and 7 in HBsAg−/HBcAb+. IL-17 inhibitor: brodalumab, ixekizumab, and secukinumab; IL-12/23 inhibitor: ustekinumab; IL-23 inhibitor: risankizumab. Abbreviations: ALT, alanine aminotransferase; bDMARDs, biologic disease-modifying antirheumatic drugs; IL, interleukin; HBV, hepatitis B virus; HBVr: hepatitis B virus reactivation; HBsAg, HBV surface antigen; HBcAb, HBV core antibody; HCV, hepatitis C virus; TNF-a, tumor necrosis factor-alpha.

**Table 2 viruses-17-00042-t002:** Analysis of the pooled incidence of HBVr in psoriasis patients treated with cytokine inhibitors.

Patient Group	No. of Records	No. of Patients	Incidence Rate (%)	95% CI	*I*^2^ (%)
HBsAg+ without antiviral prophylaxis	5	25	21.2	9.4–41.0	0
HBsAg+ with antiviral prophylaxis	6	31	0	0	-
HBsAg−/HBcAb+ without prophylaxis	12	218	4.4	2.2–8.7	0
Subgroup of HBsAg−/HBcAb+ without prophylaxis					
Drug catalog					
Interleukin 12/23 inhibitor	7	117	4.9	1.9–11.8	0
Interleukin 17 inhibitor	3	84	3.3	1.0–10.8	0
Interleukin 23 inhibitor	2	17	6.3	0.9–34.3	0
Study region					
Asian	7	142	4.5	1.9–10.6	0
Non-Asian	5	76	4.3	1.4–12.5	0
HBsAb status					
HBsAb+	10	118	6.0	2.6–13.4	0
HBsAb−	7	31	14.0	5.5–31.4	0

CI: confidence interval; HBVr: hepatitis B virus reactivation; HBsAb: hepatitis B surface antibody; HBsAg: hepatitis B surface antigen; HBcAb: hepatitis B core antibody.

## Data Availability

Data are unavailable due to privacy or ethical restrictions.

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
