# Peer review of "Incidence of HBV Reactivation in Psoriasis Patients Undergoing Cytokine Inhibitor Therapy: A Single-Center Study and Systematic Review with a Meta-Analysis"

_viruses, 2024, doi:10.3390/v17010042_

Round 1
Reviewer 1 Report
Comments and Suggestions for Authors
The authors conducted an interesting study that presents useful results. The method is clearly explained, and the results are also presented in a structured manner.
My comments are:
In the introduction section, the authors need to include recent data on the incidence of hepatitis B virus infection in patients with psoriasis.
The discussion section also needs to be improved. The discussion section is difficult to follow. I believe it should be better organized. As the reader progresses, it is often unclear whether the reference is to the current study or a previous one. This distinction should be made explicit. For example, lines 233-234 introduce a statement abruptly, and it is unclear which meta-analysis is being referred to. Similarly, lines 244-245 are unclear.
I think the authors should briefly present the data from references 4, 8, 9, and 10 as well.
Author Response
A reply to comments of the reviewer #1
The authors conducted an interesting study that presents useful results. The method is clearly explained, and the results are also presented in a structured manner.
My comments are:
Comment 1: In the introduction section, the authors need to include recent data on the incidence of hepatitis B virus infection in patients with psoriasis.
Response 1:
- We thank the reviewer for the valuable suggestion to include recent data on epidemiology of HBV among psoriasis patients. Recent studies show that the prevalence of HBV among psoriasis patients ranges from 0.45% to 5.6%, with risks varying by geographic location and population characteristics [1, 2]. This issue is particularly significant in HBV-endemic regions such as Taiwan, where the overall seropositivity rates for HBsAg and HBcAb are reported to be 4.05% and 21.3%, respectively [3].
- Accordingly, we have updated the Introduction section to include the following: (Page 1, Line 41-43)
" HBV infection among psoriasis patients ranges from 0.45% to 5.6%, influenced by geographic and population characteristics. In HBV-endemic regions such as Taiwan, the risk of reactivation represents a significant clinical challenge.”
Newly cited reference
- Suh, H. Y.; Yoon, Y. B.; Ahn, J. Y.; Park, M. Y.; Youn, J. I., Association of Hepatitis B Virus Infection and Psoriasis. Ann Dermatol 2017, 29, (6), 822-824.
- Arafa, A.; Mostafa, A., Association of hepatitis B virus infection and psoriasis: A meta-analysis. Australas J Dermatol 2020, 61, (4), 382-384.
- Chang, K. C.; Chang, M. H.; Chen, H. L.; Cheng, F. W.; Wu, J. F.; Su, W. J.; Hsu, H. Y.; Ni, Y. H., Survey of hepatitis B virus infection status after 35 years of universal vaccination implementation in Taiwan. Liver Int 2024, 44, (8), 2054-2062.
Comment 2: The discussion section also needs to be improved. The discussion section is difficult to follow. I believe it should be better organized. As the reader progresses, it is often unclear whether the reference is to the current study or a previous one. This distinction should be made explicit. For example, lines 233-234 introduce a statement abruptly, and it is unclear which meta-analysis is being referred to. Similarly, lines 244-245 are unclear.
Response 2:
- Thank you for pointing out these issues. We have restructured the discussion section to enhance its clarity and flow. The revised discussion explicitly distinguishes between findings from the current study and those from previous studies.
- Lines 233-234 have been clarified by updating the discussion section as follows: (Page 8, Line 266-277)
“A meta-analysis conducted by Papatheodoridis et al, included six studies involving rheumatologic HBsAg+ patients treated with cytokine inhibitors [7]. Among patients without antiviral prophylaxis, the pooled HBVr rate was 36% (16/45; 95% CI: 23–50%; heterogeneity, p = 0.73). In contrast, no HBVr cases were observed in patients receiving prophylaxis (0/26). Similarly, a meta-analysis by Kuo et al. focused on psoriasis patients treated with cytokine inhibitors and analyzed three studies with 17 HBsAg+ patients, reporting an HBVr rate of 25% (95% CI: 10.4–49.7%) without antiviral prophylaxis [16]. Expanding upon these findings, our updated meta-analysis incorporated five studies with 56 HBsAg+ patients. We observed a 21.2% reactivation rate (4/25; 95% CI: 9.4–41.0%) in patients without prophylaxis, compared to no cases of reactivation in those receiving prophylaxis (0/31). These results strongly underscore the critical importance of prophylactic treatment in this high-risk population.”
- Lines 244-245 were rewritten as follows: (Page 8, Line 278-287)
“The reactivation rate in HBsAg−/HBcAb+ psoriasis patients was significantly lower compared to HBsAg+ individuals [16, 17]. In our retrospective cohort, the HBVr rate was 3.2%. Papatheodoridis et al. analyzed eight studies involving 235 HBsAg−/HBcAb + patients with rheumatic disease revealed a reactivation rate of 3% (95% CI 1-6%; heterogeneity, p=0.87) among cytokine inhibitor users [7]. Similarly, a meta-analysis by Kuo et al., comprising 8 studies with 138 HBsAg−/HBcAb+ psoriasis patients, reported a reactivation rate of 5.0% (95% CI 2.3–10.8%, I² = 0%) [16]. Our updated meta-analysis, pooling data from 10 studies with 218 patients, refines the risk estimate for HBVr in this group to 4.4% (95% CI: 2.2–8.7%; I² = 0%). Although the risk is lower than in HBsAg+ patients, it persists, necessitating careful monitoring.”
Comment 3: I think the authors should briefly present the data from references 4, 8, 9, and 10 as well.
Response 3:
Thanks for your comments. We briefly present the data from references 4, 8, 9, and 10 in the results section as follows: (Page 8, Line 259-265)
“In our retrospective cohort, no HBVr was observed among HBsAg+ psoriasis patients without antiviral prophylaxis (0/5), consistent with the findings of Lu et al. (0/3; treated with ustekinumab) and Qin et al. (0/2; treated with secukinumab). In contrast, Chiu et al. and Ting et al. reported reactivation rates of 28.5% (2/7) and 25% (2/8) with ustekinumab treatment. Among those receiving antiviral prophylaxis, no cases of HBVr were observed. The limited sample sizes and variability in these studies highlight the need for meta-analyses to accurately assess the true risk.”
Reviewer 2 Report
Comments and Suggestions for Authors
Dear Ms Mina Liu
Thanks for the invitation
Concerning the manuscript ID viruses - 3365714 "Incidence of HBV Reactivation in Psoriasis Patients Undergoing Cytokine Inhibitor Therapy: A Single-Center Study and Systematic Review with Meta-Analysis"
The study is well-conducted and appropriate, and I recommend its publication without modifications. The authors acknowledge the study’s limitations and address them effectively, highlighting the need for further understanding and monitoring of the issue.
The study is significant, regarding the reactivation of the Hepatitis B virus (HBV) in psoriasis patients undergoing treatment with immunosuppressants. The authors mitigated the study’s limitation of a reduced cohort size by incorporating a systematic literature review, which adds depth and context to the research. Both the study’s methodology and the systematic review are satisfactory.
Kind regards
Author Response
Comment 1: The study is well-conducted and appropriate, and I recommend its publication without modifications. The authors acknowledge the study’s limitations and address them effectively, highlighting the need for further understanding and monitoring of the issue.
The study is significant, regarding the reactivation of the Hepatitis B virus (HBV) in psoriasis patients undergoing treatment with immunosuppressants. The authors mitigated the study’s limitation of a reduced cohort size by incorporating a systematic literature review, which adds depth and context to the research. Both the study’s methodology and the systematic review are satisfactory.
Response 1:
We sincerely thank the reviewer for their positive feedback and recommendation for publication. We are pleased that our study's methodology and approach have been recognized as appropriate and significant in addressing HBV reactivation in psoriasis patients. Thank you once again for your thoughtful and supportive review.
Reviewer 3 Report
Comments and Suggestions for Authors
This manuscript is about HBV reactivation in patients with psoriasis. With the introduction of many new drugs, this paper may be clinically useful. Some minor corrections would be needed.
1. The method used to measure the HBV serum marker, the name of the measuring company, and the cut-off level of positivity should be included.
2. The Method states that HBs antibodies are measured, but the results section is less explicit about the results; HBs antibody positive and HBc antibodies may include cases of pre-existing infection. What were the results for these cases?
3. Is HBcAb+ on line 139 a correct description?
4. There is no uniformity in the use of the terms “HBsAg-positive” or “HBsAg+”, “HBcAb-positive” or “HBcAc+”. The description of the positivity and negativity of these serum markers should be standardized. In addition, HBV reactivation and HBVr should be standardized.
5. Figure 2 is not recognized in the manuscript. Correction is desired.
6. It would be better to describe in detail the pre-treatment HBV status of the two reactivated cases, including their titer if possible.
Author Response
This manuscript is about HBV reactivation in patients with psoriasis. With the introduction of many new drugs, this paper may be clinically useful. Some minor corrections would be needed.
Comment 1: The method used to measure the HBV serum marker, the name of the measuring company, and the cut-off level of positivity should be included.
Response 1:
- We sincerely appreciate the reviewer’s valuable feedback. In response, we have revised the Materials and Methods section to provide detailed descriptions of the detection methods used for assessing clinical characteristics.
- The updated methods are as follows: (Page 3, Line 108-115)
“ALT levels were measured using automated biochemical analyzers (Roche Analytics; Roche Professional Diagnostics, Penzberg, Germany). The cut-off value for the serum ALT level was 40 IU/L. Serological markers for HBV (HBsAg, HBcAb, and HBsAb) were assessed using chemiluminescence immunoassays (Abbott ARCHITECT i2000). Serum HBV DNA levels were measured using the COBAS® HBV quantitative nucleic acid test on the COBAS® 4800 System (Roche Diagnostics), with a lower limit of quantification of 5 IU/mL. These evaluations were conducted at regular intervals in accordance with clinical guidelines.”
Comment 2: The Method states that HBs antibodies are measured, but the results section is less explicit about the results; HBs antibody positive and HBc antibodies may include cases of pre-existing infection. What were the results for these cases?
Response 2:
Thank you for bringing this to our attention. We acknowledge that the Results section did not explicitly detail the outcomes for these cases. In this cohort, only patients with evidence of previous infection (seropositive for HBsAg or HBcAb) were included. Consequently, vaccinated patients (HBsAg-/HBcAb- and HBsAb+) were excluded. Among the patients who were HBsAg-/HBcAb+, 68% (42/62) were HBsAb positive, as detailed in Table 1.
To address this, we have revised the Results section as follows: (Page 4, Line 166-167)
“Among patients with HBsAg-/HBcAb+, 68% (42/62) were HBsAb+.”
Comment 3: Is HBcAb+ on line 139 a correct description?
Response 3:
Thank you for pointing this out. Upon review, we agree that the notation "HBsAg-/HBcAb+" on line 139 was incorrect. It should indeed be "HBsAg-/HBcAb-". We have corrected the error in the manuscript to ensure accuracy and consistency. (Page 4, Line 152)
Comment 4: There is no uniformity in the use of the terms “HBsAg-positive” or “HBsAg+”, “HBcAb-positive” or “HBcAc+”. The description of the positivity and negativity of these serum markers should be standardized. In addition, HBV reactivation and HBVr should be standardized.
Response 4:
Thank you for highlighting this issue. We acknowledge the inconsistencies in the terminology describing serum marker statuses and HBV reactivation. To address this, we have made the following changes:
- Standardized the terminology throughout the manuscript. "HBsAg+" and "HBsAg−" are now consistently used to describe hepatitis B surface antigen status. "HBcAb+" and "HBcAb−" are now consistently used to describe hepatitis B core antibody status.
- Standardized the abbreviation for hepatitis B virus reactivation: "HBVr" is now used uniformly throughout the manuscript in place of "HBV reactivation."
Thank you for your valuable feedback, which has helped us improve the precision and readability of our work.
Comment 5: Figure 2 is not recognized in the manuscript. Correction is desired.
Response5:
Thank you for pointing out this oversight. Upon review, we identified that Figure 2 was not included in the manuscript. We have now added Figure 2 on page 6-7 and ensured it is appropriately referenced within the text. We appreciate your correction, which has helped improve the accuracy and completeness of our manuscript.
Comment 6: It would be better to describe in detail the pre-treatment HBV status of the two reactivated cases, including their titer if possible.
Response 6:
Thank you for your comment. We agree that providing more detailed information about the pre-treatment HBV status of the two reactivated cases will enhance the clarity and depth of our findings. Both HBVr cases were HBsAg- and HBcAb+ with an undetectable baseline HBV DNA titer. The first case had negative anti-HBs levels (<10 IU/mL), while the anti-HBs titer for the second case was unavailable (as shown in Table S3).We added the following details to the Results section:
" The first case involves a female patient who was HBsAg- and HBcAb+ prior to treatment, with an undetectable HBV DNA titer and HBsAb levels of <10 IU/mL.” (Page 5, Line 187-188)
“The second case involved a male patient who was also HBsAg- and HBcAb+, with an undetectable pre-treatment HBV DNA titer. Data on his HBsAb levels was unavailable. " (Page 6, Line 196-197)
Reviewer 4 Report
Comments and Suggestions for Authors
In this study, authors investigated the incidence of HBV reactivation in psoriasis patients undergoing cytokine inhibitor therapy by a retrospective study and a systematic review and meta-analysis. There are some suggestions.
1. It is recommended that authors describe the reason why they combined this study with a systematic review and meta-analysis
2. In the Abstract, the methods should be more detailed, especially when authors conducted both a retrospective study and a systematic review and meta-analysis.
3. Line 38: Full names of HBV should be provided.
4. In the Materials and Methods, the detection methods for clinical characteristics’ indicators need to be provided.
5. In Table 1, have authors compared the differences of baseline characteristics among different groups?
Author Response
In this study, authors investigated the incidence of HBV reactivation in psoriasis patients undergoing cytokine inhibitor therapy by a retrospective study and a systematic review and meta-analysis. There are some suggestions.
Comment 1: It is recommended that authors describe the reason why they combined this study with a systematic review and meta-analysis
Response 1:
1. We thank the reviewer for their insightful comment. The combination of our retrospective study with a systematic review and meta-analysis was intended to provide a more comprehensive evaluation of the risk of HBV reactivation in psoriasis patients treated with cytokine inhibitors. Given the limited number of cases in our single-center study, integrating data from multiple studies enhanced the statistical power and generalizability of our findings. This dual approach also allowed us to contextualize our results within the broader body of evidence, thereby offering clinicians more robust and reliable data to guide clinical decision-making.
2. To clarify this rationale, we have added the following statement to the Introduction (Page 2, line 62-66):
“Due to the limited number of cases in our single-center study, pooling data from multiple studies allowed us to increase the statistical power and generalizability of our findings. To achieve this, additionally, a comprehensive analysis was performed through a systematic review and meta-analysis, pooling data from our cohort and existing research.”
3. We have also updated the limitations (Page 9, line 307-311):
“First, the small sample size in our single-center study is a challenge. To address this, we combined our cohort data with existing research through a systematic review and meta-analysis. This combined approach enhanced the statistical power of our findings, providing clinicians with more reliable insights into HBVr risks among psoriasis patients treated with cytokine inhibitors.”
Comment 2: In the Abstract, the methods should be more detailed, especially when authors conducted both a retrospective study and a systematic review and meta-analysis.
Response 2:
We thank the reviewer for this suggestion. We have revised the Abstract to include more detailed descriptions of the methods.(Page 1, line 16-20)
“Methods: A retrospective chart review was conducted to assess the risk of HBVr in 73 psoriasis patients treated with cytokine inhibitors between 2013 and 2023. Additionally, a systematic review and meta-analysis were performed, pooling data from 10 studies (including our cohort) and adhering to PRISMA guidelines. Statistical heterogeneity was assessed using the I² statistic, and pooled proportions were calculated using a random-effects model.”
Comment 3: Line 38: Full names of HBV should be provided.
Response 3:
Thank you for pointing this out. We have updated line 38 to include the full name of HBV as "hepatitis B virus" upon its first mention to ensure clarity for all readers. (Page1, line 38)
Comment 4: In the Materials and Methods, the detection methods for clinical characteristics’ indicators need to be provided.
Response 4:
We appreciate the reviewer’s valuable feedback. In response, we have updated the Materials and Methods section to include detailed descriptions of the detection methods used for clinical characteristics. (Page 3, line 108-115)
“ALT levels were measured using automated biochemical analyzers (Roche Analytics; Roche Professional Diagnostics, Penzberg, Germany). The cut-off value for the serum ALT level was 40 IU/L. Serological markers for HBV (HBsAg, HBcAb, and HBsAb) were assessed using chemiluminescence immunoassays (Abbott ARCHITECT i2000). Serum HBV DNA levels were measured using the COBAS® HBV quantitative nucleic acid test on the COBAS® 4800 System (Roche Diagnostics), with a lower limit of quantification of 5 IU/mL. These evaluations were conducted at regular intervals in accordance with clinical guidelines.”
Comment 5: In Table 1, have authors compared the differences of baseline characteristics among different groups?
Response 5:
1. Thank you for this valuable suggestion. We have now compared the baseline characteristics between HBsAg+ and HBsAg-/HBcAb+ groups. The p-values for these comparisons were added to Table 1, using the Mann-Whitney U test for continuous variables and the Chi-square test or Fisher’s exact test for categorical variables. (Page 5, Line 174-175)
2. This information has been incorporated into the table legend and the Results section (Page 4, Line 168-172)
“Baseline characteristics were also compared between HBsAg+ and HBsAg-/HBcAb+ groups, as indicated in Table 1. Statistically significant differences were observed in baseline HBsAb status (P < 0.001) and HBV prophylaxis rates (P < 0.001). Other variables did not show statistically significant differences between the groups.”
Round 2
Reviewer 4 Report
Comments and Suggestions for Authors
Authors have addressed all issues.